# Corpus callosum anatomical changes in Alzheimer patients and the effect of acetylcholinesterase inhibitors on corpus callosum morphometry

**Ramada R. Khasawneh**[1]*, **Ejlal Abu-El-Rub**[1], **Ayman Alzu'bi**[1], **Gamal T. Abdelhady**[1,2], **Hana S. Al-Soudi**[3]

1 Faculty of Medicine, Department of Basic Medical Sciences, Yarmouk University, Irbid, Jordan, 2 Faculty of Medicine, Department of Anatomy, Ain Shams University, Cairo, Egypt, 3 Nuclear Medicine, King Hussein Medical Center, Royal Medical Services, Amman, Jordan

* Ramada@yu.edu.jo

**Data Availability Statement:** All relevant data are within the paper and its Supporting Information files.

## Abstract

The Corpus Callosum (CC) is an important structure that includes the majority of fibers connecting the two brain hemispheres. Several neurodegenerative diseases may alter CC size and morphology leading to its atrophy and malfunction which may play a role in the pathological manifestations found in these diseases. The purpose of the current study is to determine any possible changes in CC size in patients suffering from Alzheimer's disease. The Study also investigated the effect of acetylcholinesterase inhibitors (AChEIs) on the size of CC and its association with improvement in the Alzheimer disease severity scores. Midsagittal size of CC were recorded prospectively from 439 routine T1-weighted MRI brain images in normal individuals. The internal skull surface was measured to calculate CC/ internal skull surface ratio. Two groups of patients were studied: 300 (150 male / 150 female) were healthy subjects and 130 (55 males / 75 females) had Alzheimer disease. Out of the 130 Alzheimer disease pateints, 70 patients were treated with Donepezil or Rivastigmine or both. The size of the CC was measured based on T1-weighted MRI images after the treatment to investigate any possible improvement in CC size. The mean surface area of CC in controls was $6.53\pm1.105$ $cm^2$. There was no significant difference between males and females ($P < 0.627$), and CC/ internal skull surface ratio was $4.41\pm0.77\%$. Patients with mild or severe Alzheimer disease showed a significant reduction in CC size compared to healthy controls. Treating mild Alzheimer patients with either Donepezil or Rivastigmine exerts a comparable therapeutic effect in improving the CC size. There was more improvement in the size of CC in patients with severe Alzheimer disease by using combined therapy of Donepezil and Rivastigmine than using single a medication. we measured the mean size of the various portions of the corpus callosum in normal individuals and Alzheimer patients before and after taking Donepezil and Rivastigmine. Alzheimer patients have pronounced reduction in CC which is corrected after taking Donepezil and Rivastigmine leading to remarkable improvement in Alzheimer disease severity scores.

**Funding:** The author(s) received no specific funding for this work.

**Competing interests:** NO authors have competing interests

## Introduction

The Corpus Callosum (CC) is the largest commissural tract in the nervous system. CC connects similar cortical regions in the two cerebral hemispheres. Functionally, it provides interhemispheric transfer of sensory, motor, and cognitive information [1]. The midsagittal CC area (CCA) is used as an estimator of the number of small diameter fibers involved in higher-order cognitive functions [2] and a larger CCA has been hypothesized to reflect improved interhemispheric communication [3].

The CC is a prominent band of compact white matter which is composed of transversely oriented nerve fibers through which every part of each hemisphere is connected with the corresponding part of the other hemisphere. It comprises four parts: (1) the reflected anterior portion or the rostrum; (2) the genu or the anterior bulbar end; (3) the splenium or the posterior rounded end; and (4) the body which lies between the genu and the splenium. The anterior half of human CC (genu, rostrum, body) contains fibers interconnecting frontal association cortical areas, while the isthmus mostly contains a primary motor, somatosensory, and auditory fibers. Large diameter fibers are condensed in the isthmus and the posterior splenium, whereas small fibers are copious in the genu and anterior splenium.

The CC plays a crucial role in eye movement and vision by connecting the separate halves of the visual field to process images separately in each hemisphere. It also allows us to recognize the identity of objects that we visualize by connecting the visual cortex with the language centers of the brain [4]. Additionally, CC processes the tactile information in the parietal lobes and transmits the signals between the brain hemispheres to identify the nature of tactual and tangible localization. [5]. Moreover, the CC helps in maintaining the balance of arousal and attention [6]. Any disease affecting the CC is manifested by interhemispheric disconnection symptoms which include visual, auditory, kinesthetic, and complex functional impairment [7]. The CC is generally examined by brain magnetic resonance imaging (MRI), particularly its intersection with the midsagittal plane. Changes in the CC morphometry are presented in different neurodegenerative diseases which are primarily diagnosed using midsagittal brain MRI. A remarkable increase in the CC area (CCA) was observed in children affected with autistic range disorders [8], while smaller CCAs were found in schizophrenic patients [9]. Regional CC atrophy was noticed in patients with Huntington's dementia [10] and those with bipolar disorders [11]. Moreover, callosal changes were found during different stages of human development and aging [12,13]. Gender is one of the main factors responsible for callosal morphology hemispheric asymmetries. [14]. Importantly, it has been reported that a reduction in microstructural integrity of the CC, mainly in the areas connecting parietal and temporal cortices contributing to long-term working memory deficits in children suffering from traumatic brain injury.

MRI is a noninvasive imaging technology that produces detailed three-dimensional anatomical images. The introduction of MRI has permitted the evaluation of CC abnormalities *in vivo*. The employment of MRI techniques in the diagnosis of a wide range of neurologic and psychiatric diseases increased the necessity of comprehending the brain morphology in normal subjects, and comparing these normal findings with the data collected from various diseases to understand the critical influence of these changes from clinical and functional prospective. Congenital CC deformities such as agenesis, dysgenesis, and hypoplasia are the most frequent callosal anomalies, and these malformations have been reported extensively using MRI technology [15].

Several studies have been carried out to study the size and shape of the CC in different ethnic groups, including Caucasian, [16,17] Japanese [18], Indians [19], and Iranians [20]. Based on the findings of these studies, corpus callosum dimensions appear to vary in different ethnic

and racial populations; therefore, determining corpus callosum dimensions and sex-related differences is vital in the diagnosis of CC-related diseases [21].

Few studies examined the size of CC in Alzheimer disease patients, but they reported conflicting findings. For example, some studies reported that atrophy occurs early in the dementing process [22], while other studies indicated that CC atrophy occurs only in late Alzheimer disease stages [23], and further studies indicate that CC remains unchanged in patients with Alzheimer disease [24]. In the present study, we investigated the CC size using the MRI mid-sagittal imaging technique and compared these findings to what has been corroborated in other similar studies. Furthermore, we compared the CC size measurements between healthy participants and patients suffering from Alzheimer's disease. To do so, we evaluated the possible atrophy of the CC by comparing the mean surfaces of CC and the CC/ internal skull surface ratios measured on the midsagittal MR plane. CC/ internal skull surface ratio measurement was seleced because some people with a small skull may show a small corpus callosum [25]. To control the variations in skull size, we measured the areas of the midline internal skull surface by manually tracing the line through the inner table as these measurements are easier to obtain. Donepezil and Rivastigmine are acetylcholinesterase inhibitors (AChEIs) which are commonly used to manage moderate to severe cases of Alzheimer. Both medications exert their therapeutic effect through inhibiting the breakdown of acetylcholine which is a ubiquitous and widely distributed neurotransmitter associated with memory potentiation. The effect of Donepezil and Rivastigmine on CC size has not been evaluated yet. Herein, we investigate the possibility of improving CC size in Alzheimer patients who are on Donepezil or Rivastigmine treatment regimen and explore if this is associated with a reduction in Alzheimer severity.

## Materials and methods

### Sample study

This retrospective study was approved by the institutional research board at Jordan University of Science and Technology (IRB #5/134/2020).

The study participants include patients who have been referred to King Abdallah Hospital for brain MRI imaging for past five years (from 2016–2021). Beside its role as a terchary hospital, King Abdallah Hospital is consider as a training and medical educational facility, so the patients must sign a consent form that there data from their medical records can be used for a research purpose.

Individuals with a history of fractures in the skull, surgeries in the brain or skull, mass or head injury were excluded from the study. Moreover, patients with space-occupying diseases involving corpus callosum or not, such as tumors, aneurysms, and arteriovenous malformations were also excluded from the study.

The total study sample size is 430 right-handed individuals with ages ranging between 50 and 83 years. The study participants were divided into the following groups: 300 healthy control (150 males and 150 females), and 130 patients with Alzheimer disease (55 males and 75 females).

Among the 130 Alzheimer disease patients, 91 patients were diagnosed with a mild disease with clinical dementia rating (CDR) = 0.5 and 1 and mini-mental state examination (MMSE) scores between 13 to 20, while 39 participants were diagnosed with severe Alzheimer disease with a CDR = 2 and 3 and MMSE score less than 12.

The number of Alzheimer patients who were on either Donepezil or Rivastigmine or both medications medications was 70. Out of those 70 Alzheimer patients, 23 were taking Rivastigmine (3–12 mg/day), 24 were taking donepezil (5–10 mg/day), and 23 were taking both

medications. Both Donepezil and Rivastigmine treatment groups had mild and severe Alzheimr cases, while the combined treatment group had only severe Alzheimer cases. The duration of treatment was between 2 to 4 years. Those patients, who underwent the treatment, were followed up by performing regular MRI imaging. Among the 70 Alzheimer disease patients, 31 were diagnosed with a mild disease, and 39 participants were diagnosed with a severe disease.

The determination of whether the enrolled patients are healthy control or with Alzheimer disease was based solely on the clinical findings and diagnosis reported in the files of patients. For the normal individuals, the MRI examinations were performed in order to find those who were possibly associated with cerebral pathology like facial palsy, balance disorders, and scalp midline cyst or mass without any cerebral abnormality.

We excluded any other potential causes of dementia such as medications-related dementia or psychiatric disorders when selecting the Alzheimer disease cases.

## MRI imaging measurements

T1 weight images of the sagittal plane were acquired for all studied individuals using a Siemens 3.0T MRI machines scanner. The scanning protocol parameters were: repetition time = 3200 ms, echo time = 499 ms, slice thickness = 1 mm, field of view = 256 mm, 256 X 160 matrix).

Observations were recorded in the DICOM files using the manufacture's software. The corpus callosum size was evaluated by two trained radiology specialists who were blinded to the study design, sample groups, and the treatment medication. We used a fully automated method to find the midsagittal plane of the MRI [26]. The CC measurements were made on the midsagittal T1-weighted sections in each participant through areas within irregular regions of interest or with a cursor to determine four major diameters; (Fig 1) the surface area of the corpus callosum, the midline internal skull surface which includes the inner table, foramen magnum, clivus, sellar diaphragm, jugum sphenoidale, the length of CC from the anterior-most part of the genu to the posterior-most part of the splenium; and the thickness of genu, corpus, and splenium.

Hyperostosis frontalis interna measurements were not reported because they spare the midline and has a negligible effect on the midline of the internal skull surface measurements.

The genu was defined as the anterior third, the mid-body of the CC was part of the middle third, and the posterior third was subdivided into the splenium at the posterior fifth and the isthmus, a region between the mid body and the splenium.

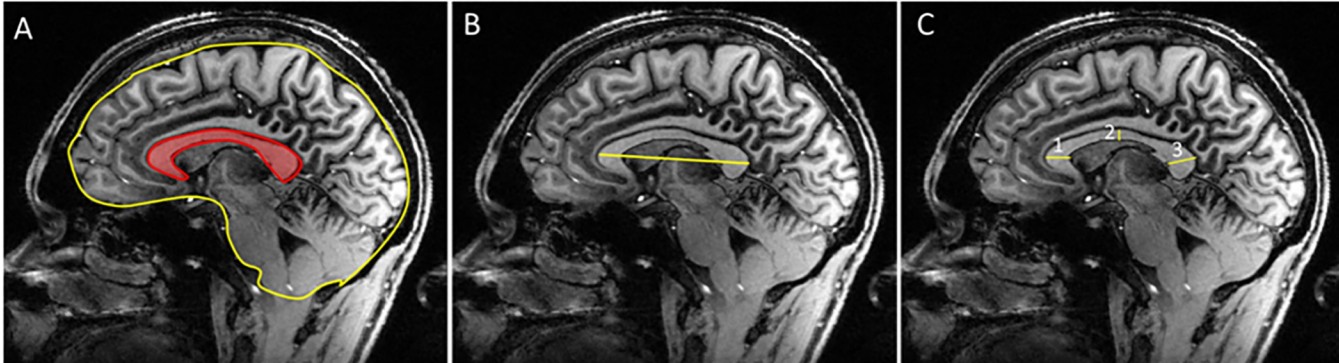

**Fig 1. Delineation and partitioning of the midsagittal corpus callosum in T1-weighted MR images.** The corpus callosum was traced in the three most medial brain slices. A, CC area is in red and whole skull surface is in yellow and B, length of the CC; C, CC thickness at genu, body, and splenium.

## Statistical analysis

The Student's *t*-test for unpaired samples was then calculated to determine if significant differences were present between healthy and diseased subsets. Each subgroup turned out to be homogeneous for sex and age ratios, respectively, by chi-squared and Student's *t*-tests. 1- For each gender and age group, Average values for CC normal surface, diameters, and CC/ internal skull surface ratio were calculated for normal volunteers and patients with normal clinical and MRI evaluations. A preliminary validation that normal patients and volunteers offered similar values had to be obtained before further investigations. 2- Overall determination of significant differences between normal females and males, and for sex and age groups (one-way and two-way ANOVA tests followed by Bonferroni's test), were performed. 3- Comparison between normal and pathologic subsets was conducted: I) For an available comparison, a subset of normal participants statistically identical for age and sex to each pathologic group was preselected by the following criteria: mean age (one-way and two-way ANOVA tests followed by Bonferroni's); sex ratio (one-way and two-way ANOVA tests followed by Bonferroni's); ii) From these data, significant differences were calculated for CC surface and CC/internal surface skull ratio. Statistical significance was tested at the level of $P = 0.05$. The data are presented as mean ± standard error of the mean (SEM).

## Results

### Reduced corpus callosum size in Alzheimer patients is associated with poor disease scores

A total of 300 normal participants were included in the study. They were divided into two groups according to sex; males (n = 150) and females (n = 150). There were no differences in CC size between males and females (Table 1). Investigating the different parts of CC in both sexes did not show any significant difference in the mean thickness of the genu, body, or splenium between both groups (Table 1). The results clearly showed that the mean thickness of splenium was no significantly larger in females, whereas the mean internal skull surface was no significantly larger in males. The results did not show any significant difference in CC/ internal skull surface ratio between males and females (Table 1).

The results of associated MRI images showed that the mean CC midsagittal surface area was 6.53±1.105 cm³. Mean diameters for the genu, body, and splenium were 9.66±1.175, 6.06 ±2.51, and 11.01±1.35 cm, respectively. The length of the CC was 7.3±0.31 cm and the mean

**Table 1. Average CC mean values normal healthy individuals.**

|  | Total subjects N = 300 | Males N = 150 | Females N = 150 | Significant |
|---|---|---|---|---|
| Mean age ± SE | 66.155±10.05 | 66.24±10.4 | 66.07±9.7 | Non sig |
| Mean CC surface ± SE (cm) | 6.53±1.105 | 6.47±1.2 | 6.6±1.01 | Non Sig |
| Mean genu thickness ± SE (mm) | 9.66±1.175 | 9.78±1.1 | 9.54±1.25 | Non Sig |
| Mean Body thickness ± SE (mm) | 6.06±2.51 | 6.34±1.41 | 6.24±1.1 | Non Sig |
| Mean splenium thickness ± SE (mm) | 11.01±1.35 | 10.87±1.2 | 11.15±1.5 | Non Sig |
| Length of CC ± SE (cm) | 7.3±0.31 | 7.32±.24 | 7.28±0.38 | Non Sig |
| Mean internal skull surface ± SE (cm²) | 148.06±12.325 | 149.68±12.8 | 146.44±11.85 | Non Sig |
| CC/internal skull surface ratio ± SE (%) | 4.41±0.77 | 4.322±0.29 | 4.506±0.67 | Non Sig |

No significant ($P > 0.05$). Determination of control subsets homogeneous for age and sex by chi-squared test.

internal skull surface of normal participants was 148.06±12.325 cm$^2$ (Table 1). The CC ratio to the brain was 4.41±0.77% (Table 1).

There was no difference in CC measurements associated with the age or gender of the normal participants. Since no statistical difference was found in these measurements, we validate the appropriateness to exclude the age and gender as a factors that could affect the CC size, whereas several recently conduced studies have failed to identify significant CC sex and age differences [27–29]. The absence of differences between males and females are summarized in Table 1 which included the mean surface of the CC, mean body thickness of the CC, the mean internal skull surface, and the CC/internal skull surface ratio.

All above reported results lead to the conclusion that the age and gender risk factors don't affect the CC surface measurements or the CC/internal skull surface ratio in healthy participants. The CC size measurements in normal participants were used as a platform to compare it with any possible changes in CC size measurements in Alzheimer patients.

The study group for Alzheimer disease included 55 males and 75 females with an average age of 65 years. Out of the 130 Alzheimer disease patients, 91 were diagnosed with mild Alzheimer and 39 patients were diagnosed with severe Alzheimer disease.

The measurements showed a significant decrease in the CC size in Alzheimer disease patients compared with the normal control group. The mean thicknesses of the genu, body, and the splenium were significantly reduced in Alzheimer patients compared to normal control. The CC/internal skull surface ratio was also significantly lesser in Alzheimer patients compared to normal control (Table 2).

A comparison of CC size between males and females with Alzheimer disease was investigated. The results showed a significant decrease in total CC size in Alzheimer male patients compared to female ones. However, our data showed that the mean body thickness was significantly smaller in males compared to females with Alzheimer disease. Moreover, the mean thickness of genu and splenium was not affected in both groups. The CC/internal skull surface ratio was also calculated; the data did not show any significant difference in CC/internal skull surface ratio between males and females with Alzheimer disease (Table 3).

To investigate the relationship between the severity of the Alzheimer disease and the CC size, Alzheimer patients were divided into two subgroups; the first group included mild Alzheimer patients, and the second group included those with severe Alzheimer disease. Our results show that severe cases of Alzheimer disease have more pronounced reduction in the CC size compared to those with mild Alzheimer. Moreover, the CC / brain ratio significantly decreased in severe Alzheimer patients compared to mild Alzheimer patients (Fig 2). The

**Table 2. Average CC values in Alzheimer disease individuals: Comparison with normal values.**

|  | Control | Alzheimer disease | Significance |
|---|---|---|---|
| Mean age ± SE | 66.155±10.05 | 65.09±5.75 | Non Sig |
| Mean CC surface ± SE (cm) | 6.53±1.105 | 5.415±1.33 | $P \leq 0.05$ |
| Mean genu thickness ± SE (mm) | 9.66±1.175 | 7.855±0.175 | $P \leq 0.05$ |
| Mean Body thickness ± SE (mm) | 6.06±2.51 | 4.55±1.345 | $P \leq 0.05$ |
| Mean splenium thickness ± SE (mm) | 11.01±1.35 | 9.3±1.275 | $P \leq 0.05$ |
| Length of CC ± SE (cm) | 7.3±0.31 | 7.04±1.025 | Non Sig |
| Mean internal skull surface ± SE (cm$^2$) | 148.06±12.325 | 145.16±10.425 | Non Sig |
| CC/internal skull surface ratio ± SE (%) | 4.41±0.77 | 3.7±0.24 | $P \leq 0.05$ |

No significant ($P > 0.05$). Determination of control subsets homogeneous for age and sex by chi-squared test.

**Table 3. Average CC values in Alzheimer disease patients.**

|  | Total subjects N = 130 | Males N = 55 | Females N = 75 | Significance |
|---|---|---|---|---|
| Mean age ± SE | 65.09±5.75 | 65.88±5.4 | 64.30±6.1 | Non Sig |
| Mean CC surface ± SE (cm) | 5.415±1.33 | 5.11±1.12 | 5.72±1.54 | $P \leq 0.05$ |
| Mean genu thickness ± SE (mm) | 7.855±0.175 | 7.9 ± 0.24 | 7.81 ± 0.11 | Non Sig |
| Mean Body thickness ± SE (mm) | 4.55±1.345 | 4.2±1.28 | 4.9±1.41 | $P \leq 0.05$ |
| Mean splenium thickness ± SE (mm) | 9.3±1.275 | 9.4 ± 1.24 | 9.2 ± 1.31 | Non Sig |
| Length of CC ± SE (cm) | 7.04±1.025 | 7.14±0.95 | 6.94±1.1 | Non Sig |
| Mean internal skull surface ± SE (cm²) | 145.16±10.425 | 146.9±10.35 | 143.42±10.5 | Non Sig |
| CC/internal skull surface ratio ± SE (%) | 3.7±0.24 | 3.4±0.19 | 3.9 ± 0.3 | Non Sig |

No significant ($P > 0.05$). Determination of control subsets homogeneous for age and sex by chi-squared test.

internal skull surface did not change between the control patients, mild Alzheimer, and severe Alzheimer patients (Fig 2).

These findings clearly proved the correlation between the size of CC and the severity scores of Alzheimer disease at the time of diagnosis.

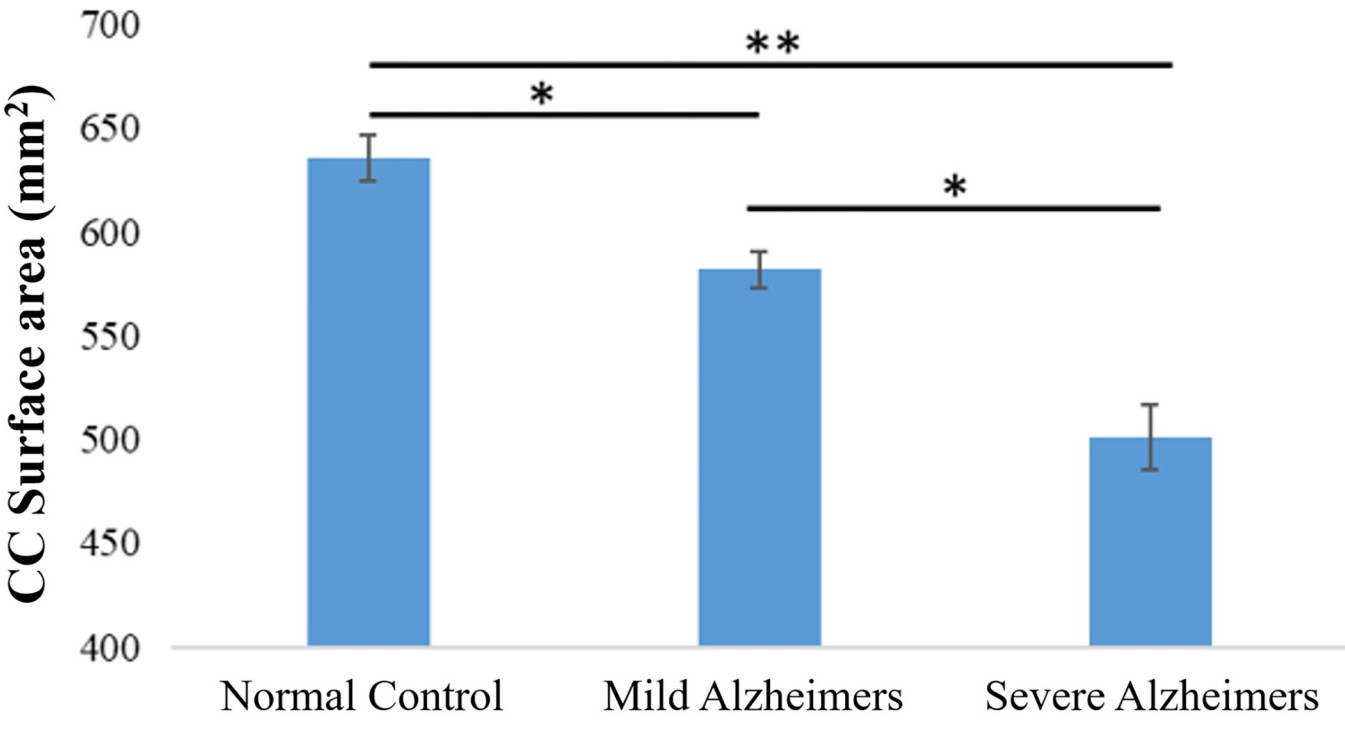

**Fig 2. Adjusted corpus callosum CC mean values for the three diagnostic groups; normal controls, mild Alzheimer patients, and severe Alzheimer patients.** Corpus callosum (CC) mean values were significantly smaller in Alzheimer patients compared to normal controls. Moreover, the CC area and size decreases as the severity of Alzheimer increases, thus severe Alzheimer patients have significant reduction in the CC size compared to mild Alzheimer patients. Each column represents the mean CC area ± standard error of the mean (SE). *P<0.05, **<0.01 (t-test).

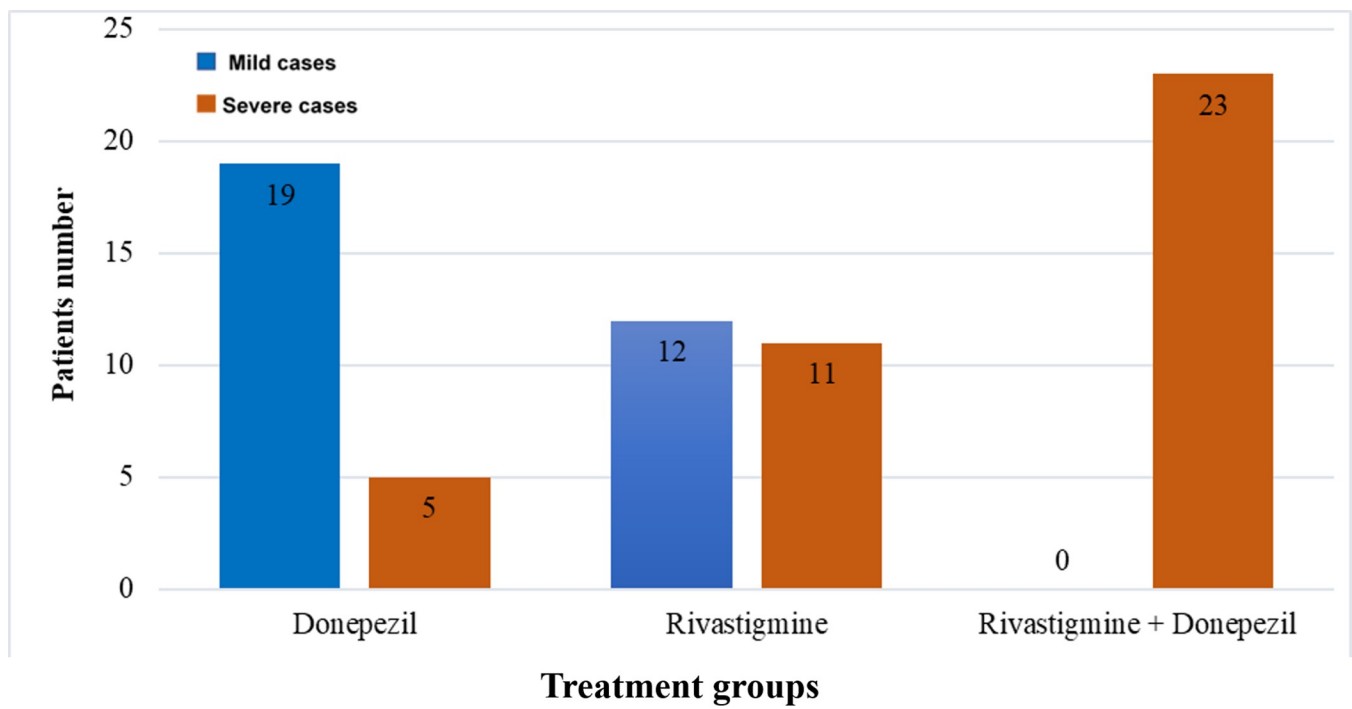

**Fig 3. Descriptive data for the treated Alzheimer patients.**

### Corpus Callosum size increased in Alzheimer patients after taking Donepezil or Rivastigmine medications resulting in improving the severity scores and delaying the progression of the Alzheimer disease

Many Alzheimer patients were commonly prescribed acetylcholinesterase inhibitors (AChEIs) such as Donepezil, Rivastigmine, or combination of both to slow the cognitive impairments in Alzheimer patients and improve the severity scores. No previous study has investigated the effect of acetylcholinesterase inhibitors (AChEIs)- Donepezil or Rivastigmine- on improving the cognitive and memory functions of Alzheimer patients by targeting the CC. So we examine whether Alzheimer disease patients have an improvement in CC size after being treated with Donepezil or Rivastigmine or both. We assessed 70 patients with clinical diagnosis of AD, 24 patients were treated using donepezil, 23 patients were treated using Rivastigmine, while the remaining 23 patients were treated using both Donepezil and Rivastigmine (Fig 3). The duration of treatment and the medications used were shown in S1 Table.

Herein, we assessed the interrelation between taking Donepezil or Rivastigmine or both medications and the improvement in the CC size and Alzheimer disease severity scorers -MMSE score and CDR- (S1 Table). The degree of CC atrophy was measured by MRI imaging before and after using the Alzheimer medications.

MRI imaging was performed for the patients to calculate the CC size before and after the treatment of choice. Furthermore, our results showed no significant difference between Donepezil treatment group and Rivastigmine treatment group in the degree of improvement in CC size and MMSE, and CDR severity scores indicated that both medications are therapeutically equivalent in enhancing the CC size and improving the cognitive and memory functions in mild or severe cases of Alzheimer disease (Figs 4 and 5, Table 5).

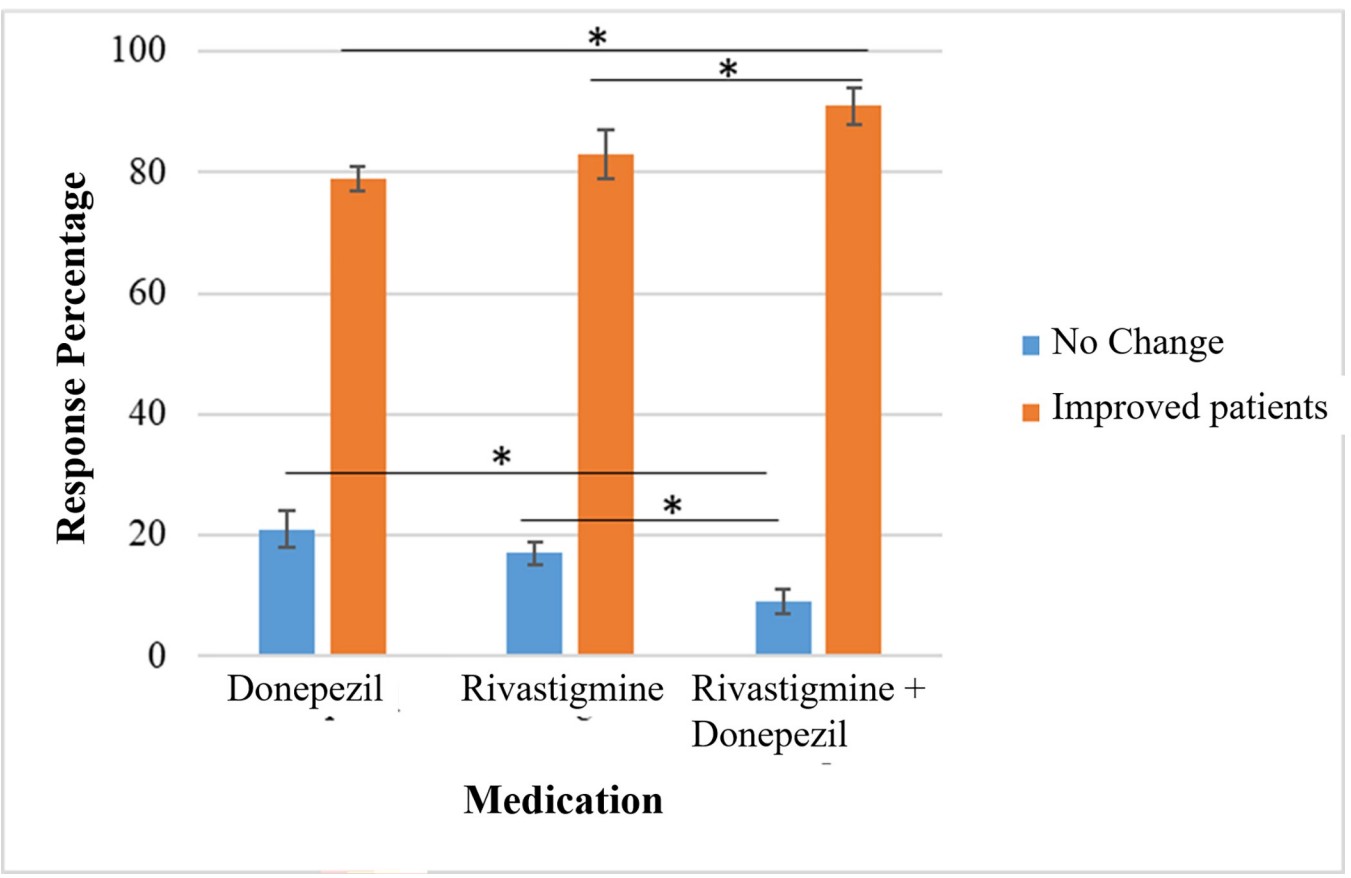

**Fig 4. Percentage of improvement in Alzheimer severity scores after acetylcholinesterase inhibitors (AChEIs) medications were given to the patients.** No significant difference reported between Donepezil treatment group and Rivastigmine treatment group in the degree of improvement in severity scores. On the other hand, severe cases of Alzheimer showed a more significant increase after using a Donepezil and Rivastigmine combined therapy than using a single medication. Each column represents the percentage of severity score improvement ± standard error of the mean (SE). *$P < 0.05$ (*t*-test).

Interestingly, our results revealed that severe cases of Alzheimer showed significant increase in the CC size and a more noticeable improvement in MMSE and SDR scores after using a combined therapy with Donepezil and Rivastigmine than using single medication (Figs 4–6, Tables 4 and 5). The results of the current study highlighted for the first time the correlation between CC size and the severity of Alzheimer disease, and the possibility of increasing the CC size and improving the cognitive functions in Alzheimer patients by prescribing acetylcholinesterase inhibitors (AChEIs). These results provide new insights about the pathological scenarios that can lead to the delineated novel therapeutic role of acetylcholinesterase inhibitors (AChEIs)- Donepezil and Rivastigmine- in Alzheimer patients.

## Discussion

The size of CC has variably been evaluated *in vivo* using MRI either in normal individuals or in pathologic conditions [14,18,30]. There is much disagreement in the literature about whether or not specific parts of the Corpus Callosum show sexual dimorphism. Our results show that with or without considering the brain size, there are no sex-specific differences in the size of the corpus callosum, so these results are consistent with what has been published before [21,31–34].

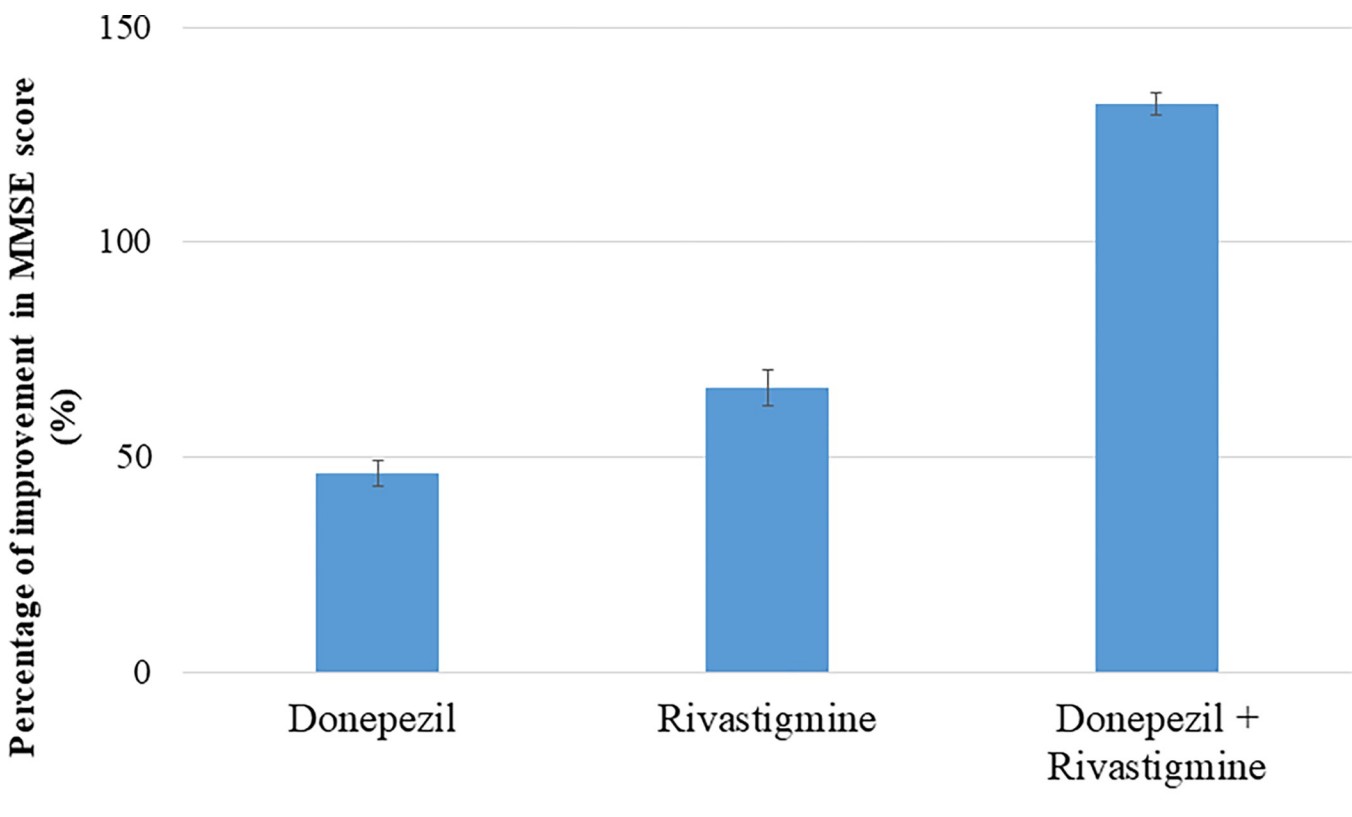

**Fig 5. Percentage of improvement in MMSE score with each medication.** The best improvement in MMSE score was seen when using a combined therapy with donepezil and rivastigmine in contrast with a single medication.

As these results are inconsistent with Bermudez and Zatorre's study, they showed that the total area of CC was significantly larger in men because they have anterior third and posterior midbody [16]. Moreover, Ardekani *et al*. (2013) showed that the average CC was significantly larger in females [35].

Based on the severity of the disease, patients with Alzheimer disease experience variable degrees of cognitive and memory impairments. The relationship between the severity of

**Table 4. Average CC values in mild Alzheimer disease patients: Before and after the treatment.**

|  | Mild patients before treatment | Mild patients after treatments | Significance |
|---|---|---|---|
| **Mean CC surface (cm)** | 5.88±1.42 | 6.21±1.68 | $P \leq 0.05$ |
| **Mean genu thickness (mm)** | 8.3±0.165 | 8.91±0.219 | $P \leq 0.05$ |
| **Mean Body thickness (mm)** | 5.13±1.329 | 6 ±1.621 | $P \leq 0.05$ |
| **Mean splenium thickness (mm)** | 10 ±1.3 | 10.88±1.31 | $P \leq 0.05$ |
| **Length of CC (cm)** | 7.15±1.1 | 7.2±1.08 | Non Sig |
| **Mean internal skull surface (cm²)** | 147±11.297 | 147±12.49 | Non Sig |
| **CC/internal skull surface ratio (%)** | 4±0.24 | 4.22±0.48 | $P \leq 0.05$ |

Paired sample *t*-tests.

No significant ($P > 0.05$).

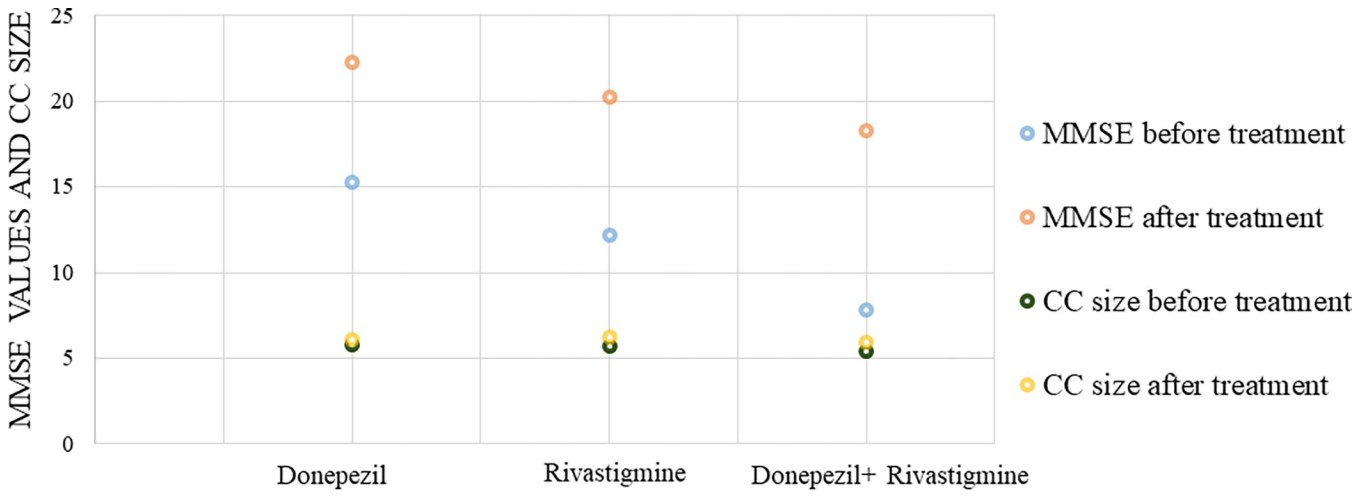

**Fig 6. A scatter plot show the mean CC improvement versus mean MMSE score improvement for each medication.** The scotter plot showed that MMSE score improvement was assocaiated with increase in CC size, moreover, using a using combined therapy with donepezil and rivastigmine gives a better improvement in both MMSE score and CC size compared with a single medication.

Alzheimer disease and the CC size has not been extensively investigated. Since CC involved in orchestrating many cognitive and memory functions in the central nervous system (CNS), there is a possibility that Alzheimer disease pathological mechanisms may be associated with changes in CC size and morphology. The total CC area, the thickness of CC subregions, and the length of CC were all measured and compared in age and matched Alzheimer disease patients. Our results demonstrate that the measurements of CC are substantially different in mild and severe Alzheimer disease patients compared to control patients. Our results reveal a significant decrease in the size of CC in Alzheimer's patients compared to control patients. Moreover, the difference of decrease in CC size between the normal controls and mild Alzheimer patients is found to be significant. Also, there is a difference of decrease in the size of CC between mild and severe Alzheimer patients, and such a result indicated that as the Alzheimer severity increased, the degree of reduction in the CC size also increased.

Furthermore, our show that Alzheimer's disease is manifested by remarkable CC atrophy with (19%) reduction in the total CC area in comparison to control patients. These results are consistent with previous studies that reported a significant correlation between CC atrophy

**Table 5. Average CC values in severe Alzheimer disease individuals: Before and after treatments.**

|  | Severe patients before treatment | Severe patients after Treatment | Significant |
|---|---|---|---|
| Mean CC surface (cm) | 5.62±1.42 | 6.09±1.68 | $P \leq 0.05$ |
| Mean genu thickness (mm) | 7.5±0.165 | 8.1±0.219 | $P \leq 0.05$ |
| Mean Body thickness (mm) | 4.15±1.329 | 5.45±1.621 | $P \leq 0.05$ |
| Mean splenium thickness (mm) | 9.1±1.3 | 10±1.31 | $P \leq 0.05$ |
| Length of CC (cm) | 7.1±1.1 | 7.16±1.08 | Non Sig |
| Mean internal skull surface (cm$^2$) | 147±7.34 | 147±9.86 | Non Sig |
| CC/internal skull surface ratio (%) | 3.8±0.24 | 4.14±0.48 | $P \leq 0.05$ |

Paired sample $t$-tests.

No significant ($P > 0.05$).

and Alzheimer's disease [9,23,36,37]. The age risk factor in this study was matched between disease and control patients, and it is important to exclude the possibility of age-related CC atrophy that occurs in the normal aging process. [38]. Moreover, our data delineated that the total CC area of Alzheimer-diseased females (5.72±1.54 cm$^2$) was significantly larger than Alzheimer-diseased males (5.11±1.12 cm$^2$) of the same age and disease severity. A similar gender-related difference in the CC area was also observed in the healthy controls which confirms that the rating and severity of CC atrophy in Alzheimer disease patients are not associated with gender variance [35]. Many studies consistently reported a noticeable reduction in the total CC area in Alzheimer disease patients compared with healthy controls. However, few inconclusive studies focused on reporting findings related to the CC regional atrophy in Alzheimer patients. These studies reported various and inconsistent results regarding the region of CC atrophy found in Alzheimer patients. For example, Thomann *et al.* reported the presence of CC atrophy in the anterior portion only [39], whereas Frederiksen *et al.* and Wang *et al.* reported that CC atrophy merely occurs in the posterior portion [9,22]. Several other studies showed that the CC atrophy can be found in both anterior and posterior regions in Alzheimer disease patients [23,36,40,41], while Hensel *et al.* found no specific regional atrophy, but overall CC atrophy in Alzheimer disease patients [42]. These variations in regional CC atrophy may be due to several factors, including normal variation in the morphology of CC which depends on the age, gender, brain size, the handedness [12,35,43,44], the size of study samples [9,23,41], the methodological approaches adopted in defining and measuring the CC subregions [45], and finally discrepancies in the stage of Alzheimer's disease in patients recruited in these studies [37].

Notably, in our study, we examined the thickness of CC in different parts (genu, body, and splenium). Our results show a considerable reduction in thickness of all parts that are composed of the CC area in patients with Alzheimer's disease weighed against healthy controls. The most prominent reduction was observed in the body (25%), followed by the genu (18%), while the least observed reduction was in splenium (15%). These results indicate for the first time that the reduction in the total CC area in Alzheimer disease patients is not only occurring in specific CC subregion as previously reported [9,22,39], but could be a result of collective involvement of all CC subregions despite the difference in the degree of atrophy between the different subregions.

Alzheimer disease arises from the degeneration of the intricate neural network in cerebral cortex and hippocampus [46]. The lost cortical neurons initiated by the progressive decline in the concentrations of acetylcholine neurotransmitter which is crucial to main intact Cortical neuronal network involved in memory, attention, learning, and other cognitive processes [47]. Since the CC is included in the cerebral cortex, it will be significantly impaired and changed as a result of low concentrations of acetylcholine, but the exact consequences have been inconclusively studied. In our study, we demonstrate a dramatic decrease in the size of CC in Alzheimer disease patients. These findings prompted us to find out whether the commonly used medications for Alzheimer disease can restore the size of CC and improve the cognitive functions in Alzheimer disease patients. Donepezil is an acetylcholinesterase inhibitor that helps in preventing the degradation of acetylcholine and increasing its concentration, which may be useful in enhancing memory and cognitive functions through preserving the remaining neurons in the cerebral cortex and Hippocampus [48,49]. Rivastigmine is another acetylcholinesterase inhibitor, but it also inhibits Butyrylcholinesterase enzyme [48]. Butyrylcholinesterase takes over and continues the acetylcholine degradation process when acetylcholinesterase is lost, thus Rivastigmine is superior to Donepezil in increasing the concentration of acetylcholine [48]. The effect of both medications on CC size has not been investigated yet. Our study is the first to investigate the efficacy of Donepezil and Rivastigmine in restoring the size of CC.

Patients with mild Alzheimer can benefit from a single therapy with either Donepezil or Rivastigmine, which provides an equal effect in restoring the decline in the CC size. Patients with severe Alzheimer showed more improvement in the size of CC by using combined therapy of Donepezil and Rivastigmine. These findings provide new avenues to further understand the therapeutic effects of these acetylcholinesterase inhibitors. Future studies should be designed to investigate the mechanistic pathway exerted by Donepezil and Rivastigmine to improve and restore the size of CC in Alzheimer patients, we can anticipate that Donepezil and Rivastigmine can potentiate the neuro-regeneration process that will increase the neuronal network in the CC and improve the memory and cognitive functions in Alzheimer patients that lead to better prognosis. CC is responsible for interhemispheric correlations that can modulate and control many higher brain functions such as the cognition [50]. Since these acetylcholinesterase inhibitors reduce the breakdown of Acetylcholine and boost its level in the neuronal cells resulting in enhancing the synaptic transmission and restore the integrity of neuronal network in the CC [51]. It has been shown that the loss of Acetylcholine can disturb the integrity of CC and initiate its degeneration. Further studies revealed that the presence of Acetylcholine in plentiful amount could act on the M2 muscarinic receptors of neural stem cells, thus increasing their proliferation, differentiation and recruitment which may help in restoring the nerve fibers network in the CC [52]. The significant correlation between the reduction in the overall CC area thickness and the progression in the severity of Alzheimer disease is in agreement with previous studies which reported that the thickness profile of CC is considered as an important indicator for the progression of many neurodegenerative pathologies [53]. Furthermore, continuous monitoring of the CC regional thickness is clinically fundamental to predicting the possibility of transition from mild cognitive impairment to a severe form of Alzheimer's disease [37,54]. Additionally, the therapeutic efficacy of Alzheimer medications can be assessed by observing the changes in the CC size. Our findings set up a clinically important platform by illustrating the importance of continuous observation and examination of CC thickness as a prognostic tool in patients with AD and other related neurodegenerative disorders. Furthermore, the CC thickness changes can be used to monitor the effectiveness of prescribed medications for AD patients. Moreover, the findings of the current study can be used to study the possible correlation between these medications and the morphometry of other structures that control the memory and cognition mainly hippocampus.

## Supporting information

**S1 Table. Clinical characteristics of patients with Alzheimer disease including the disease severity scores before and after treatments.**
(DOCX)

## Acknowledgments

We would like to thank Mr. Muhammad Abu El-Rub for the time and effort spent in reviewing the Manuscript

## Author Contributions

**Conceptualization:** Ramada R. Khasawneh, Ejlal Abu-El-Rub.

**Data curation:** Hana S. Al-Soudi.

**Formal analysis:** Ramada R. Khasawneh.

**Investigation:** Ramada R. Khasawneh.

**Methodology:** Ramada R. Khasawneh.

**Writing – original draft:** Ramada R. Khasawneh, Ejlal Abu-El-Rub, Ayman Alzu'bi, Gamal T. Abdelhady.

**Writing – review & editing:** Ramada R. Khasawneh, Ejlal Abu-El-Rub.

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
