## [Decision Letter · Decision Letter 0]

8 Mar 2022

PONE-D-22-04511Corpus Callosum anatomical changes in Alzheimer patients and the effect of acetylcholinesterase inhibitors on Corpus Callosum morphometryPLOS ONE

Dear Dr. Khasawneh,

Thank you for submitting your manuscript to PLOS ONE. After careful consideration, we feel that it has merit but does not fully meet PLOS ONE’s publication criteria as it currently stands. Therefore, we invite you to submit a revised version of the manuscript that addresses the points raised during the review process.

We look forward to receiving your revised manuscript.

Kind regards,

Yasmina Abd‐Elhakim

Academic Editor

PLOS ONE

Journal Requirements:

"NO, The funders had no role in study design, data collection and analysis, decision to publish, or preparation of the manuscript."

**Reviewers' comments:**

**Reviewer #1:** S.No. Saction Line no. Remarks

1. Methods Explain how the sample size was calculated.

2. Methods 139, 140 The total study sample size is 439. The study participants written were divided into

the groups: 300 healthy control and 130 patients with Alzheimer disease. So total no.

of participants were 130+300=430. What about remaining 9 patients.

3. Methods 141-144 Explain briefly about clinical dementia rating (CDR) and mini-mental state

examination (MMSE) scores and criteria for differentiation into mild & severe

Alzheimer disease.

Authors have taken into account only mild & severe cases. What about moderate

cases.

4. Methods 145-149 Since the study is retrospective, methodology is written in present tense. Need

corrections.

5. Methods 158-159 Authors are requested to explain that how did they differentiate between

medications-related dementia & mild Alzheimer disease.

6. Results 225-227 Authors validate the appropriateness to exclude the age and gender themself as a

factors that could affect the CC size but the sample size is too small in the present

study to validate.

Authors are requested to confirm validation with reference to previous studies.

7. Results

Discussion 278 – 287, 415-416 Authors should comment upon clinical outcome of the disease after medications

and side effects if any due to chronic use of ace inhibitors.

8. Discussion 345 – 349 Irrelevant, can be omitted

**Reviewer #2:** This manuscript presents corpus callosum (CC) measurements for a large number of normal human subjects (150 males; 150 females) in comparison to comparable measurements from 130 Alzheimer’s Disease patients (55 males; 75 females). The authors report no sex differences for the controls, and significant reductions in CC measurements for the patient population. For a large subset of the patient population (n=70), the authors also report improvements in MMSE scores and increases in CC measurements associated with treatment with Donepezil, Rivastigmine, or a combination of both drugs. The authors have a quite substantial body of data, and the findings are clearly worthy of publication. That said, there are (mostly minor) issues that should be addressed.

1) It is always impressive to see authors for whom English is not their primary language develop manuscripts. This manuscript is in desperate need of editorial assistance from someone with greater proficiency in English.

2) The authors generally need to exercise more caution in describing their results. See below.

3) Line 138: the authors need to correct a typographical error – not 439, but 430

4) Line 178: for part A, CC area is in red and whole skull surface is in yellow (rather than the reverse as stated in the legend)

5) Line 204: the authors state that “The CC was larger in females than males…” This is not so, as clearly shown in Table 1 where non-significant differences are reported for all comparisons. So, one does not say, “it was larger but not significant.” One says “there were no differences.”

6) Lines 227-228: Similarly, Table 1 does not present “differences” between males and females; rather the absence of differences

7) Lines 251-252: The authors state that “The results show a significant decrease to male patients in total CC size in Alzheimer females compared to male patients.” This is not what the numbers in Table 1 say (5.11 for males; 5.72 for females.

8) Lines 252-253; Similarly, mean body thickness was not significantly smaller in females…

9) Lines 300-301: The authors state they find “no significant differences between Donepezil…and Rivastigmine…in the degree of improvement in CC size and MMSE…” Table 4 does not distinguish the drugs.

10) In examining the data presented in Supplementary Table 1, this reviewer cannot work out what exactly the “Response Percentage” axis signifies in Figure 4. I see only one patient (# 59) who showed no improvement in MMSE score. Indeed, the before and after MMSE scores presenting in the supplementary table are impressive. Why not simply graph % improvement in MMSE score with drug treatment.

11) Moreover, since the authors have MMSE and CDR scores and CC measures for the drug-treated patients, it would be of interest to see scatter plots of CC improvement versus MMSE score improvement, along with some simple correlation coefficients. The authors certainly imply a relationship between these measures, and they have the data to show the extent to which a relationship exists.

12) Lines 345-346: The authors state that “in this study, we found that women had a larger splenium than men.” They report no significant difference in Table 1.

13) Lines 369-371: The authors state that the total CC area of Alzheimer females was higher than males (yes, but Table 3 values would suggest that it is all due to differences in mean body thickness). They go on to state “A similar gender-related difference in …healthy controls.” This is not so (see Table 1).

14) The finding that CC sizes increase with drug treatment is remarkable. The authors might wish to speculate on the cause of this change. Seemingly, it would have to be to either increases in the number of fibers (which would see unlikely) or an increase in the caliber of the fibers. Are there ultrastructural data in the literature?

---

## [Author Response · Author response to Decision Letter 0]

18 Apr 2022

Point-by-point response:

Your valuable comments, suggestions, and remarks are highly valued and appreciated, and they will be taken into consideration.

Dear Editor:

1. The manuscript meets PLOS ONE's style requirements. 

2. we also attached the ethical approval as requested.

3. The authors received no specific funding for this work.

4. regarding the consent form, King Abdallah Hospital is consider as a training and medical educational facility, so the patients must sign a consent form that there data from their medical records can be used for a research purpose .. this sentence was added to the text 

Reviewers' comments:

Reviewer #1

1. Methods 139, 140 The total study sample size is 439. The study participants written were divided into the groups: 300 healthy control and 130 patients with Alzheimer disease. So total no. of participants was 130+300=430. What about remaining 9 patients?

Response: Thank you for your notification. It was a typo mistake, we already corrected the total number of participants to be 430. 

2. Methods 141-144 Explain briefly about clinical dementia rating (CDR) and mini-mental state examination (MMSE) scores and criteria for differentiation into mild & severe Alzheimer disease. Authors have taken into account only mild & severe cases. What about moderate cases.

Response: Thank you for your valuable comment. We are taking in consideration the cases that are mild and severe where the treatment can either slow disease progression and slow the cognitive impairment as in mild cases or restore some cognitive functions that are hugely deteriorated as in severe cases. It is difficult to find moderate cases since the majority of Alzheimer’s patients are either stays for longer time in early or severe stages while the moderate or middle-stage is so evanescent and usually considered a short term stage. As mentioned in the paper, for mild Alzheimer patients: clinical dementia rating (CDR)was 0.5 and 1 and mini-mental state examination (MMSE) scores were between 13 to 20, while for severe Alzheimer patients the CDR was 2 and 3 and MMSE score was less than 12.

3. Methods 145-149 Since the study is retrospective, methodology is written in present tense. Need corrections

Response: Thank you for your insightful observation. We already went through the methodology part and re-wrote it in the past tense.

4. Methods 158-159 Authors are requested to explain that how did they differentiate between medications-related dementia & mild Alzheimer disease. 

Response: Dementia and Alzheimer’s disease aren’t the same. Dementia is the term applied to a group of symptoms that negatively impact memory, but Alzheimer’s is a specific progressive disease of the brain that slowly causes impairment in memory and cognitive function.

Currently, many studies suggested a stronger correlation between certain drug groups and the rise in the risk of dementia in elderly patients. For example, there were several reports that linked benadryl (diphenhydramine), antipsychotics and Parkinson's medications with dementia onset. Further, Anticholinergic drugs which help in relaxing muscles by blocking acetylcholine, an important chemical that transmits messages in the nervous system, can also induce cognitive impairment. We already excluded patients who are taking medications for psychiatric and neuro- diseases or chronically used prescribed medications.

5. Results 225-227 Authors validate the appropriateness to exclude the age and gender themself as a factors that could affect the CC size but the sample size is too small in the present study to validate. Authors are requested to confirm validation with reference to previous studies

Response: We thank the reviewer for their careful evaluation of our manuscript, three references were added to confirm the exclusion of age and gender as a factors that could affect the CC size. These references are numbered in the revised manuscript as 27, 28, and 29.

6. Discussion 278 – 287, 415-416 Authors should comment upon clinical outcome of the disease after medications and side effects if any due to chronic use of ace inhibitors 

Response: We thank you for this comment. We already mentioned in the manuscript that the improvement in CC has been associated with improvement in the clinical dementia rating (CDR) and mini-mental state examination (MMSE) scores. Since our focus to investigate the correlation between acetylcholinesterase inhibitors and CC size, we didn’t mention the side effects, further, some of the patients selected to be included in the current study are still taking these medications and no reported side effects have been found in the patients’ files. 

7. Discussion 345 – 349 Irrelevant, can be omitted

Response: We thank the reviewer for their careful evaluation of our manuscript. The sentence has been omitted

Reviewer #2

1. It is always impressive to see authors for whom English is not their primary language develop manuscripts. This manuscript is in desperate need of editorial assistance from someone with greater proficiency in English.

Response: We highly appreciated your comment. We already assigned an expert to edit and proofread the manuscript in order to correct any grammatical and spelling errors. 

2. The authors generally need to exercise more caution in describing their results. See below from number 3 to number 9:

Response: We thank the reviewer for his careful evaluation of our manuscript. All these mistakes were corrected as requested in the revised manuscript.

3. In examining the data presented in Supplementary Table 1, this reviewer cannot work out what exactly the “Response Percentage” axis signifies in Figure 4. I see only one patient (# 59) who showed no improvement in MMSE score. Indeed, the before and after MMSE scores presenting in the supplementary table are impressive. Why not simply graph % improvement in MMSE score with drug treatment?

Response: Thank you for this suggestion. We already added a graph % labeled as fig 5 to the main text, which shows the improvement in MMSE score with drug treatment 

4. Moreover, since the authors have MMSE and CDR scores and CC measures for the drug-treated patients, it would be of interest to see scatter plots of CC improvement versus MMSE score improvement, along with some simple correlation coefficients. The authors certainly imply a relationship between these measures, and they have the data to show the extent to which a relationship exists.

Response: Thank you for this suggestion. We already added a scatter plots of CC improvement versus MMSE improvement to the main text, it is labeled as fig 6.

5. Lines 345-346: The authors state that “in this study, we found that women had a larger splenium than men.” They report no significant difference in Table 1 

Response: We thank the reviewer this comment. It was an unintentional oversight, and we rephrased the sentence and corrected the mistake presented.

6. Lines 369-371: The authors state that the total CC area of Alzheimer females was higher than males (yes, but Table 3 values would suggest that it is all due to differences in mean body thickness). They go on to state “A similar gender-related difference in …healthy controls.” This is not so (see Table 1).

Response: Thank you for this comment. The mean internal skull surface is almost constant between males and females with Alzheimer’s disease, which means that the significant difference in the body thickness lead to significant decrease in CC size in comparing to the constant internal skull surface as shown by the statistical results.

7. The finding that CC sizes increase with drug treatment is remarkable. The authors might wish to speculate on the cause of this change. Seemingly, it would have to be to either increases in the number of fibers (which would see unlikely) or an increase in the caliber of the fibers. Are there ultrastructural data in the literature?

Response: We highly appreciate this suggestion. Actually the improvement in CC size with medications is an intriguing outcome to be investigated in future studies. We can anticipate possible mechanisms for this improvement. CC is responsible of interhemispheric correlations that can modulate and control many higher brain functions such as the cognition. Since these acetylcholinesterase inhibitors reduce the breakdown of Acetylcholine and boost its level in the neuronal cells, this can enhance the synaptic transmission and restore the integrity of neuronal network in the CC. It has been shown that the loss of Acetylcholine can disturb the integrity of CC and initiate its degeneration. Further, studies revealed that the presence of Acetylcholine in plentiful amount can act on the M2 muscarinic receptors of neural stem cells, thus increasing their proliferation, differentiation and recruitment which may help in restoring the nerve fibers network in the CC. The findings of the current study can be used to design future studies to rule out the possible mechanisms that lead to the improvement in CC size, and also these findings can be used to study the possible correlation between these medications and the morphometry of other structures that control the memory and cognition mainly hippocampus.

---

## [Decision Letter · Decision Letter 1]

1 May 2022

PONE-D-22-04511R1Corpus Callosum anatomical changes in Alzheimer patients and the effect of acetylcholinesterase inhibitors on Corpus Callosum morphometryPLOS ONE

Dear Dr. Khasawneh,

Thank you for submitting your manuscript to PLOS ONE. After careful consideration, we feel that it has merit but does not fully meet PLOS ONE’s publication criteria as it currently stands. Therefore, we invite you to submit a revised version of the manuscript that addresses the points raised during the review process.

We look forward to receiving your revised manuscript.

Kind regards,

Yasmina Abd‐Elhakim

Academic Editor

PLOS ONE

Journal Requirements:

Reviewers' comments:

Reviewer's Responses to Questions

**Comments to the Author**

1. If the authors have adequately addressed your comments raised in a previous round of review and you feel that this manuscript is now acceptable for publication, you may indicate that here to bypass the “Comments to the Author” section, enter your conflict of interest statement in the “Confidential to Editor” section, and submit your "Accept" recommendation.

Reviewer #1: All comments have been addressed

Reviewer #2: All comments have been addressed

2. Is the manuscript technically sound, and do the data support the conclusions?

Reviewer #1: Yes

Reviewer #2: Yes

3. Has the statistical analysis been performed appropriately and rigorously? 

Reviewer #1: Yes

Reviewer #2: Yes

4. Have the authors made all data underlying the findings in their manuscript fully available?

Reviewer #1: (No Response)

Reviewer #2: Yes

5. Is the manuscript presented in an intelligible fashion and written in standard English?

Reviewer #1: Yes

Reviewer #2: Yes

6. Review Comments to the Author

**Reviewer #1: **Authors have written that This retrospective study was approved by the institutional research board. Authors didn't mention about institutional ethical committee approval.

**Reviewer #2:** The authors have adequately addressed all of the issues raised in the initial review. They have added figures to flesh out the presentation of their results.

7. PLOS authors have the option to publish the peer review history of their article (what does this mean?). If published, this will include your full peer review and any attached files.

Reviewer #1: No

Reviewer #2: **Yes: **Preston E Garraghty

---

## [Author Response · Author response to Decision Letter 1]

13 May 2022

Point-by-point response:

Your valuable comments, suggestions, and remarks are highly valued and appreciated, and they will be taken into consideration

Journal Requirements:

1. Please review your reference list to ensure that it is complete and correct. If you have cited papers that have been retracted, please include the rationale for doing so in the manuscript text, or remove these references and replace them with relevant current references. Any changes to the reference list should be mentioned in the rebuttal letter that accompanies your revised manuscript. If you need to cite a retracted article, indicate the article’s retracted status in the References list and also include a citation and full reference for the retraction notice

Response: Dear Editor, we have checked all the references, I would like to assure you that all references are complete and correct, and we do not have any retracted references.

Reviewer #1 

1. Authors have written that This retrospective study was approved by the institutional research board. Authors didn't mention about institutional ethical committee approval

Response: We thank the reviewer for his careful evaluation of our manuscript, the ethical approval was claimed from the human ethical committee in the Jordan University of Science and Technology, the ethical approval number was IRB #5/134/2020, this already mentioned and highlighted in the manuscript, we also attached the ethical approval document which is in Arabic and a translated copy in English.

Reviewer #2

We would like to thank Prof. Preston E Garraghty for his participation in reviewing and evaluating this paper, which help improving the manuscript, thank you.

---

## [Editor Report · Decision Letter 2]

16 May 2022

Corpus Callosum anatomical changes in Alzheimer patients and the effect of acetylcholinesterase inhibitors on Corpus Callosum morphometry

PONE-D-22-04511R2

Dear Dr. Khasawneh,

We’re pleased to inform you that your manuscript has been judged scientifically suitable for publication and will be formally accepted for publication once it meets all outstanding technical requirements.

Kind regards,

Yasmina Abd‐Elhakim

Academic Editor

PLOS ONE
---

## [Editor Report · Acceptance letter]

26 May 2022

PONE-D-22-04511R2 

Corpus Callosum anatomical changes in Alzheimer patients and the effect of acetylcholinesterase inhibitors on Corpus Callosum morphometry 

Dear Dr. Khasawneh:

I'm pleased to inform you that your manuscript has been deemed suitable for publication in PLOS ONE. Congratulations! Your manuscript is now with our production department. 

Kind regards, 

on behalf of

Dr. Yasmina Abd‐Elhakim 

Academic Editor

PLOS ONE